# Positive Association of Serum Alkaline Phosphatase Level with Severe Knee Osteoarthritis: A Nationwide Population-Based Study

**DOI:** 10.3390/diagnostics10121016

**Published:** 2020-11-27

**Authors:** Hye-Min Park, Jun-Hyuk Lee, Yong-Jae Lee

**Affiliations:** 1Department of Family Medicine, Yonsei University College of Medicine, 211 Eonju-ro, Gangnam-gu, Seoul 06273, Korea; jadore-hm@yuhs.ac; 2Department of Medicine, Graduate School, Yonsei University, 50-1 Yonsei-ro, Seodaemoon-gu, Seoul 03722, Korea; muzzyljh@yuhs.ac; 3Department of Family Medicine, Yongin Severance Hospital, Yonsei University College of Medicine, 363 Dongbaekjukjeon-daero, Giheung-gu, Yongin-si, Gyeonggi-do 16995, Korea

**Keywords:** serum alkaline phosphatase, severe knee osteoarthritis, low-grade inflammation

## Abstract

Serum alkaline phosphatase (ALP), a well-known marker of hepatobiliary and bone disorders, has recently been discovered to be a biochemical marker of cardiometabolic diseases and chronic low-grade inflammation. We aimed to evaluate the association of serum ALP level with knee osteoarthritis in the general population. The study included 3060 men and women aged ≥50 years who participated in the 2009–2011 Korea National Health and Nutrition Examination Survey. The participants were categorized into three groups based on log-transformed serum ALP level as follows: T1 (1.74–2.32), T2 (2.33–2.43), and T3 (2.44–3.01). Their radiographs were evaluated by two well-trained radiologists using the Kellgren–Lawrence (KL) grading system. After excluding those with KL Grade 0, we categorized the remaining participants into two groups, a severe osteoarthritis group (KL Grade 4) and a non-severe osteoarthritis group (KL Grades 1 to 3). The odds ratios (ORs) with 95% confidence intervals (CIs) of severe osteoarthritis according to the tertiles of log-transformed serum ALP levels of patients with osteoarthritis were calculated using a weighted multivariate logistic regression analysis. Compared with T1, the adjusted ORs (95% CIs) for severe osteoarthritis of the T3 serum ALP group was 1.613 (1.087–2.394; *p* = 0.018) after adjusting for the confounding variables. Conclusively, serum ALP activity was independently and positively associated with severe knee osteoarthritis in middle-aged and older adults.

## 1. Introduction

Osteoarthritis is the most common type of arthritis in old adults, with a substantial cost to the individual and society [1,2]. The Centers for Disease Control and Prevention reported that 240 million people worldwide have osteoarthritis, including >30 million people in the United States, an increase from 21 million in 1990 to 27 million in 2005. In South Korea, the prevalence of knee osteoarthritis has been estimated to be 4.5% among men and 19% among women according to the 2010–2012 Fifth Korean National Health and Nutrition Examination Survey (KNHANES-V) [3]. While the development and progression of osteoarthritis vary between individuals, most people have joint pain and functional limitations [4].

In the past, osteoarthritis was considered a result of aging and identified as a joint degenerative disease [5]. However, the pathophysiological mechanisms of osteoarthritis are still under discussion. More recently, the multifactorial risk factors of osteoarthritis, such as genetic factors, obesity, and subclinical inflammation, have been associated with the progressive phases of cartilage degeneration [6]. The severity of osteoarthritis commonly ranges from an asymptomatic, incidental finding from a radiological examination to advanced incapacitating disorder [4]. The severe cartilage loss in osteoarthritis leaves “joint failure” with disability and deteriorated quality of life, and contributes to mortality [3]. Alkaline phosphatase (ALP) is primarily derived from the bones, liver, and placenta. Abnormal ALP levels in blood have long been regarded to be associated with liver, gall bladder, and bone disorders. However, emerging evidence suggests that serum ALP level is considered an inflammatory mediator associated with cardiometabolic diseases, including metabolic syndrome, type 2 diabetes, hypertension, and dyslipidemia [7,8]. Moreover, other osteoarthritis-predictive inflammatory markers such as C-reactive protein (CRP) level and leukocyte count increased with increasing serum ALP levels [9,10,11,12]. In recent years, accumulated evidence shows that systemic low-grade inflammation, which reflects metabolic overload, plays an important role in the pathogenesis and progression of osteoarthritis [13,14].

In light of these novel findings, we aimed to investigate the relationship between serum ALP levels and knee osteoarthritis in a Korean population aged ≥50 years, using data from the 2009 to 2011 KNHANES.

## 2. Materials and Methods

### 2.1. Study Population

All the data used in this study were obtained from the 2009 to 2011 Korean National Health and Nutrition Examination Survey. The KNHANES, a nationwide representative survey conducted by the Korea Centers for Disease Control and Prevention (KCDC), has a cross-sectional, multistage, stratified, probability sampling design based on sex, age, and geographical area. To capture a sample representing the general Korean population, participants are assigned sample weights. The detailed methods associated with the KNHANES have been described previously [15].

Among the 7759 adults aged ≥50 years with knee radiographs who participated in the 2009–2011 KNHANES, we applied the following exclusion criteria: (1) no abnormal findings on knee radiographs (*n* = 2792), (2) missing data on serum ALP level (*n* = 403), (3) positive for hepatitis B viral surface antigen (*n* = 149), (4) heavy alcohol consumer (*n* = 211), (5) patients with rheumatoid arthritis (*n* = 147), (6) patients with osteoporosis (*n* = 456), or (7) presence of osteoporosis on dual energy X-ray absorptiometry (*n* = 541). Finally, 3060 participants were included in the study (Figure 1). Written informed consent was provided to each participant by the KCDC during the survey. The use of KNHANES data was approved by the Institutional Review Board of the Korea Centers for Disease Control and Prevention (IRB No. 2009-01CON-03-2C, 2010-02CON-21-C, 2011-02CON-06-C). The data used in this study are available for free on the KNHANES website (http://knhanes.cdc.go.kr) for academic research purposes.

### 2.2. Biochemical Measurements

After at least 8 h of fasting, blood samples were collected from the antecubital vein from each participant. Serum total cholesterol, vitamin D, aspartate aminotransferase (AST), alanine aminotransferase (ALT), ALP, and plasma glucose concentrations were measured using a Hitachi 7600 Analyzer. We used log-transformed serum ALP level instead of serum ALP level in this study because the serum ALP levels of the participants had a skewed distribution. The participants were divided into three groups according to the tertile of log-transformed serum ALP levels as follows: T1 (1.74–2.32), T2 (2.33–2.43), and T3 (2.44–3.01).

### 2.3. Assessment of Knee Osteoarthritis

The participants underwent bilateral standing anteroposterior and lateral radiography of both knees using a SD3000 Synchro Stand (Accele Ray Shinyoung Co., Seoul, Korea). The radiographs were evaluated by two well-trained radiologists using the Kellgren–Lawrence (KL) grading system as follows [16]: Grade 0 = normal, no radiological findings of osteoarthritis; Grade 1 = doubtful of osteoarthritis, doubtful narrowing of the joint space and possible osteophytic lipping; Grade 2 = mild osteoarthritis, definite osteophytes and possible narrowing of the joint space; Grade 3 = moderate osteoarthritis, moderate multiple osteophytes, definite narrowing of the joint space, small pseudocystic areas with sclerotic walls, and possible deformity of bone contour; Grade 4 = severe osteoarthritis, large osteophytes, marked narrowing of the joint space, severe sclerosis, and definite deformity of the bone contour. After excluding those with KL Grade 0, we categorized the participants into two groups, a severe osteoarthritis group (KL Grade 4) and a non-severe osteoarthritis group (KL Grades 1 to 3). Participants with at least KL Grade 1 on their knee radiographs were asked whether they had knee pain and assessed using a numerical rating scale (NRS) ranging from 0 (no pain) to 10 (the worst pain imaginable). We categorized the NRS scores into four categories as follows: no pain, mild pain (NRS scores of 1 to 3), moderate pain (NRS scores of 4 to 6), and severe pain (NRS scores of 7 to 10).

### 2.4. Covariates

For the height and body weight measurements, the participants wore light indoor clothing without shoes, and the values were rounded off to the nearest 0.1 cm and 0.1 kg, respectively. Body mass index (BMI) was calculated as body weight divided by the square of height (kg/m^2^). Systolic and diastolic blood pressures were defined as the average of the last two of the three measured values. Among participants who responded that they smoke currently, we defined those who had smoked at least 100 cigarettes during their lifetime as current smokers [17]. Smoking status was divided into two categories, current smoker and non-current smoker. The amount of alcohol intake was calculated as the number of grams of alcohol consumed per day (g/day). After excluding heavy alcohol drinkers who drank alcohol ≥30 g/day for men and ≥20 g/day for women, we divided alcohol drinking status as social drinker or non-drinker. The International Physical Activity Questionnaire-short form was used to estimate the participants’ physical activity levels. Either vigorous activity for ≥20 min for ≥3 days/week or moderate-intensity activity for ≥30 min for ≥5 days/week were regarded as regular exercise [18,19]. We divided the participants into two groups, those with and those without regular exercise. Equivalized income was used to represent income considering household size. Monthly household income was calculated by dividing the self-reported monthly household income by the square root of the number of household members. Following this, monthly household income was classified into quartiles (lowest, mid-lower, mid-higher, and highest) [20]. We classified employment status as employed and unemployed.

For dietary surveillance, well-trained dietitians conducted in-person interviews with the participants, using the 24-h recall method. The daily amounts of calcium (Ca; mg/day) and phosphorus (P; mg/day) intakes were calculated using the data obtained through the 24-h recall method. The detailed nutrition survey protocol is presented on the KNHANES website [21].

### 2.5. Statistical Analyses

The normal distribution of the serum ALP levels was evaluated along with a determination of skewness using the Kolmogorov–Smirnov test. All data were presented as mean or percentage (%) ± standard error (SE) values. The clinical characteristics of the study population according to the tertiles of log-transformed serum ALP levels were compared using a weighted generalized linear regression analysis for continuous variables and a weighted chi-square test for categorical variables. The comparison of the mean values of the log-transformed serum ALP levels according to osteoarthritis severity was analyzed using a weighted generalized linear regression analysis. A weighted chi-square test was used to calculate the percentage of the participants with knee pain according to osteoarthritis severity. The odds ratios (ORs) with 95% confidence intervals (CIs) for severe osteoarthritis according to the tertiles of log-transformed serum ALP levels of the osteoarthritis patients were calculated using weighted multivariate logistic regression analyses. The linear trends across the tertiles of log-transformed serum ALP level were tested using the median value within each tertile as a continuous variable. All statistical analyses were performed using SPSS version 25.0 statistical software (SPSS Inc., Chicago, IL, USA). The significance level was set at *p* < 0.05.

## 3. Results

### 3.1. Clinical Characteristics of the Study Population

Table 1 presents the clinical characteristics of 3060 subjects according to the tertiles of log-transformed serum ALP levels. The mean age of the total population was 63.4 ± 0.2 years without statistical significance. The proportion of male sex, social drinker, moderate osteoarthritis, and severe osteoarthritis increased with increasing log-transformed serum ALP tertiles. The proportion of the lowest quartile of monthly household income gradually increased as log-transformed serum ALP tertiles increased. On the other hand, the proportion of the highest quartile of monthly household income gradually decreased as log-transformed serum ALP tertiles increased. Similar trends but with no statistically significant differences were observed in the proportion of current smokers, regular exercise, employed status, and knee joint pain among the groups. The mean levels of plasma glucose, serum AST, and ALT increased, and the mean vitamin D level decreased in accordance with the log-transformed serum ALP tertiles. The mean values of BMI, dietary Ca intake, dietary P intake, and dietary Ca/P intake ratio did not show statistical significance among the groups. Furthermore, as the severity of osteoarthritis worsened, percentage of obese participants and the mean value of BMI increased (Appendix A).

### 3.2. Relationship between Serum ALP Level and Severe Osteoarthritis

Figure 2 shows the mean value of the log-transformed serum ALP levels according to knee osteoarthritis severity. The mean log-transformed serum ALP levels were 2.38 and 2.41 in non-severe and severe osteoarthritis, respectively (*p* < 0.001).

Figure 3 shows the percentages of the knee pain scale scores according to the log-transformed ALP tertiles. Although the patients in the T3 groups complained of severe knee pain the most, the knee pain severity was not significantly different among the groups (*p* = 0.544). Regardless of osteoarthritis severity, the trends remained similar (*p* = 0.829 in the severe osteoarthritis subgroup and *p* = 0.551 in the non-severe osteoarthritis subgroup, respectively).

Table 2 shows the results of the weighted multivariate logistic regression model for the analysis of the relationship between serum ALP level and severe osteoarthritis. The ORs (95% CIs) for severe osteoarthritis of T3 compared with the referent T1 was 1.696 (1.182–2.433) (*p* = 0.004). Similar trends were observed even after further adjustment for all potential confounding variables, including age, sex, BMI, dietary Ca/P intake ratio, regular exercise, current smoking status, social drinking status, monthly household income, employment status, and serum vitamin D level. The fully adjusted ORs (95% CIs) for severe osteoarthritis of T3 compared with T1 was 1.613 (1.087–2.394) (*p* = 0.018). In addition, although no significant difference was found in the odds of T2 and T1, the fully adjusted ORs with 95% CIs increased according to the tertiles of log-transformed serum ALP level increments (*p* for trend = 0.017).

## 4. Discussion

In this nationally representative cross-sectional study, we found that higher serum ALP levels were positively associated with the presence of radiologically confirmed symptomatic knee osteoarthritis in middle-aged and old adults. These positive associations were maintained after adjustment for confounding variables, indicating that the elevated ALP levels are associated with a higher prevalence of severe knee osteoarthritis. The participants with higher log-transformed serum ALP level (2.44–3.01) had a 1.7-fold higher risk for the presence of severe knee osteoarthritis than the participants with lower log-transformed serum ALP level (1.74–2.32). To our knowledge, only one previous animal study has examined the relationship between synovial bone-specific ALP level and osteoarthritis in 22 horses. Fuller et al. reported a positive correlation between synovial bone-specific ALP level and cartilage damage in equine osteoarthritis [22]. Although serum ALP level has been revealed as a major biochemical marker for the evaluation of disease activity in ankylosing spondylitis and rheumatoid arthritis, we believe that this study is the first to show a positive association between serum ALP level and osteoarthritis in a nationwide general population [23,24].

The exact mechanism by which serum ALP level correlates with osteoarthritis is not fully understood, but several possible reasons for this relationship warrant consideration. First, serum ALP level could be linked to low-grade inflammation, which may cause an inflammatory response in chondrocytes. Emerging studies have demonstrated that serum ALP level is positively and independently associated with inflammatory markers such as CRP level and leukocyte counts [9,25]. In addition, increased ALP expression can be a cellular reaction to inflammatory stimuli through purinergic signaling which involves chronic inflammation. ALP plays a role in catalyzing pro-inflammatory adenosine triphosphate to anti-inflammatory adenosine. In this context, it is more likely that serum ALP is an epiphenomenon of inflammation rather than the cause of osteoarthritis [26]. Traditionally, osteoarthritis has been considered a consequence of aging and mechanical burden. However, osteoarthritis is now regarded as a complex interplay of biomechanical and cellular factors that contribute to the end-stage pathology of the joints [27]. Accumulated evidence suggests that an imbalance in metabolic conditions is a significant risk factor for the progression of osteoarthritis. The coexistence of metabolic syndrome components such as overweight, hypertension, and diabetes showed an even higher prevalence of hand osteoarthritis, which is a non-weight-bearing joint lesion [28]. Another study also found that type 2 diabetes could be an independent risk factor of osteoarthritis, showing the concept of a diabetes-induced osteoarthritis through a subclinical inflammation that has a potential impact on the progression of osteoarthritis [29]. Moreover, Hiraiwa et al. reported that hyperglycemia has harmful effects on human meniscal cells through advanced glycation end-products and reactive oxidative stress, which cause subchondral bone destruction and chondrocyte dysfunction [30]. In this study, fasting plasma glucose levels gradually increase in accordance with serum ALP quartiles. In this regard, chronic low-grade systemic inflammation may describe positive associations between them.

Second, elevated serum ALP could be associated with chronic pain. Kamimura el al. found that serum bone specific ALP were remarkably increased in elderly women with lower back pain [31]. This study showed that the participants with severe knee osteoarthritis were more likely to experience pain. Pain induced chronodisruption could be at least partly associated with severity of osteoarthritis. It is well established that persistent pain can provoke sleep disturbance, which is directly connected to alteration of the circadian rhythm. Inversely, sleep disruption can also alter pain processing and lead to a higher prevalence of pain through sensitization and habituation [32]. Chronodisruption could be linked to a substantial increase in inflammatory mediators such as CRP and tumor necrosis factor-α levels, which were increased in osteoarthritic tissue [33,34,35]. As mentioned earlier, elevated serum ALP levels were definitely associated with increased inflammatory mediators such as CRP level and leukocyte counts [9,25]. In addition, Berenbaum et al. reported that metabolic dysregulation, inflammatory pathways, and the daily rhythmic environment have all been connected to osteoarthritis susceptibility through neuroendocrine systems such as cortisol and parathyroid hormone [36].

This study has some limitations that require explanation. We could not assess degenerative disc disease, fat mass, skeletal muscle mass, synovial bone-specific ALP level, and other inflammatory biomarkers, including adipokines and chemokines due to lack of data. In addition, there was no information about patellar OA in the KNHANES. In addition, because participants only checked X-ray on knees and hips in the KNHANES, presence of osteoarthritis in other joints could affect ALP levels although there were no significant differences in serum ALP level according to the presence or severity of hip osteoarthritis (data not shown). Finally, we could not evaluate knee osteoarthritis using the Ahlbäck classification. Despite these potential limitations, our findings, using a complex sampling analysis, could be representative of the actual condition of older Korean adults with osteoarthritis.

## 5. Conclusions

Serum ALP activity was independently and positively associated with severe knee osteoarthritis, implying that serum ALP level might be a valuable additional surrogate biomarker in the evaluation of severe knee osteoarthritis.

## Figures and Tables

**Figure 1 diagnostics-10-01016-f001:**
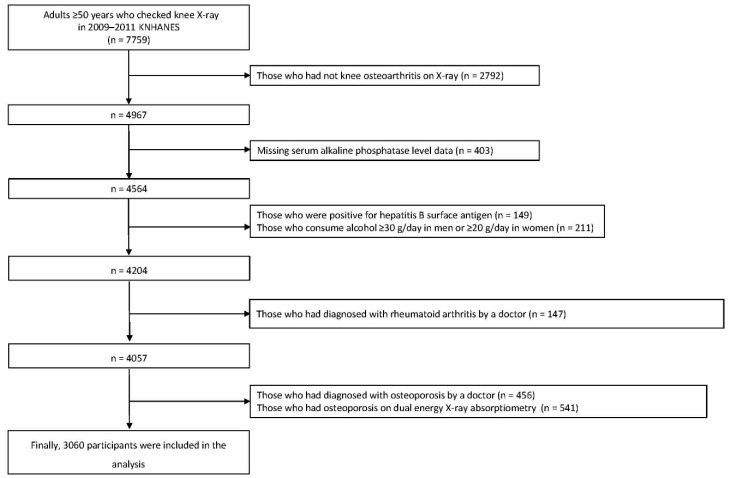
Flowchart of study population selection.

**Figure 2 diagnostics-10-01016-f002:**
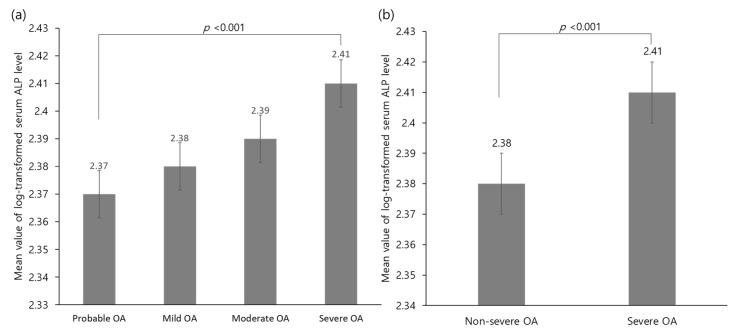
Comparison of the mean value of the log-transformed serum ALP levels according to (**a**) severity of OA or (**b**) between severe and non-severe OA. Abbreviations: ALP, alkaline phosphatase; OA, osteoarthritis.

**Figure 3 diagnostics-10-01016-f003:**
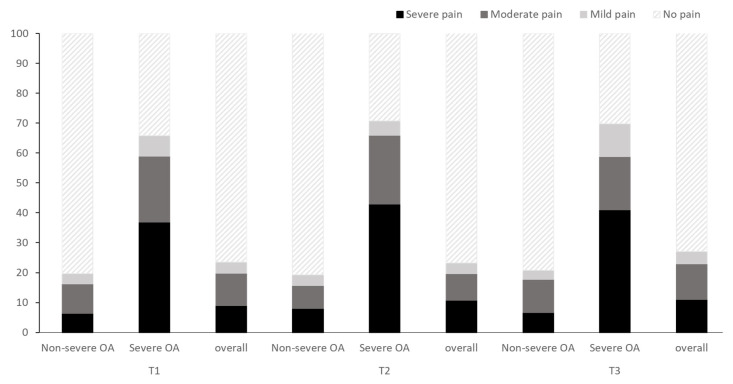
Percentage of knee pain scale scores according to the tertiles of log-transformed ALP levels. Abbreviations: ALP, alkaline phosphatase; OA, osteoarthritis.

**Table 1 diagnostics-10-01016-t001:** Clinical characteristics of the study population.

Log-Transformed Serum ALP Level	2009–2011 KNHANES				
	T1 (1.74–2.32)	T2 (2.33–2.43)	T3 (2.44–3.01)	Total	*p*
Unweighted *n*	1017	1001	1042	3060	
Male sex, % (SE)	52.8 (1.8)	50.9 (1.8)	44.0(1.9)	49.2(1.1)	0.003
Age, years	62.9 ± 0.4	63.2 ± 0.3	64.1± 0.4	63.4 ± 0.2	0.082
Body mass index, kg/m^2^	24.6 ± 0.1	24.6 ± 0.1	24.5± 0.1	24.6 ± 0.1	0.911
Dietary Ca intake, mg/day	504.4 ± 12.6	479.3 ± 12.9	485.8 ± 17.0	489.8 ± 9.2	0.289
Dietary P intake, mg/day	1131.0 ± 19.3	1122.8 ± 20.6	1076.7 ± 21.5	1110.2 ± 13.3	0.124
Dietary Ca/P ratio	0.4 ± 0.0	0.4 ± 0.0	0.4 ± 0.0	0.4 ± 0.0	0.375
Mean blood pressure, mmHg	92.8 ± 0.6	93.0 ± 0.5	94.2 ± 0.7	93.4 ± 0.3	0.218
Fasting plasma glucose, mg/dL	101.9 ± 0.7	102.6 ± 0.9	107.9 ± 1.2	104.1 ± 0.6	<0.001
Total cholesterol, mg/dL	191.7 ± 1.3	192.7 ± 1.4	194.7 ± 1.5	193.1 ± 0.8	0.293
Serum vitamin D, ng/mL	20.3 ± 0.3	19.6 ± 0.3	19.1 ± 0.3	19.7 ± 0.2	0.007
Serum AST, U/L	23.1 ± 0.3	23.4 ± 0.3	25.4 ± 0.5	24.0 ± 0.2	<0.001
Serum ALT, U/L	21.1 ± 0.5	21.7 ± 0.5	23.5 ± 0.6	22.1 ± 0.3	0.014
Current smoker, % (SE)	31.1 (2.4)	28.2 (2.0)	30.7 (2.1)	30.1 (1.2)	0.603
Current drinker, % (SE)	52.3 (2.0)	43.3 (1.9)	40.8 (1.7)	45.6 (1.1)	<0.001
Regular exercise, % (SE)	22.9 (1.8)	19.5 (1.7)	19.9 (1.7)	20.8 (1.1)	0.256
Employed status, % (SE)	59.1 (2.0)	57.2 (1.9)	54.8 (2.0)	57.1 (1.3)	0.208
Monthly household income, % (SE)					0.001
Lowest	29.5 (1.9)	30.9 (1.8)	33.9 (1.8)	31.5 (1.2)	
Medium-lowest	22.5 (1.8)	28.5 (1.6)	28.7 (1.9)	26.5 (1.1)	
Medium-highest	22.7 (1.7)	21.8 (1.7)	21.8 (1.9)	22.1 (1.2)	
Highest	25.2 (2.2)	18.8 (1.7)	15.6 (1.5)	19.9 (1.1)	
Severity of OA, % (SE)					0.002
Probable OA	46.4 (1.9)	47.8 (2.0)	39.4 (2.1)	44.5 (1.2)	
Mild OA	26.9 (1.8)	23.8 (1.7)	25.5 (1.7)	25.5 (1.1)	
Moderate OA	19.2 (1.3)	20.8 (1.4)	23.0 (1.6)	21.0 (0.9)	
Severe OA	7.5 (1.0)	7.6 (0.9)	12.0 (1.3)	9.1 (0.7)	
Knee joint pain, % (SE)					0.544
No pain	76.5 (1.9)	76.9 (1.9)	73.1 (2.0)	75.4 (1.2)	
Mild	3.7 (0.7)	3.6 (0.7)	4.0 (0.8)	3.8 (0.4)	
Moderate	10.8 (1.4)	8.9 (1.3)	12.0 (1.4)	10.6 (0.7)	
Severe	9.0 (1.1)	10.7 (1.4)	11.0 (1.3)	10.2 (0.8)	

Abbreviations: KNHANES, Korea National Health and Nutrition Examination Survey; ALP, alkaline phosphatase; Ca, calcium; P, phosphorus; AST, aspartate aminotransferase; ALT, alanine aminotransferase; OA, osteoarthritis; SE, standard error.

**Table 2 diagnostics-10-01016-t002:** Odds ratios with 95% confidence intervals for severe osteoarthritis according to the tertiles of log-transformed serum ALP level among osteoarthritis patients.

	T1	T2		T3			
		ORs (95% CIs)	*p*	ORs (95% CIs)	*p*	Overall *p*	*p* for Trend
Unadjusted	1 (ref)	1.025 (0.723–1.413)	0.889	1.696 (1.182–2.433)	0.004	0.004	0.004
Model 1	1 (ref)	1.036 (0.719–1.493)	0.848	1.528 (1.050–2.223)	0.027	0.045	0.025
Model 2	1 (ref)	1.077 (0.704–1.650)	0.731	1.620 (1.097–2.393)	0.016	0.035	0.015
Model 3	1 (ref)	1.079 (0.706–1.648)	0.726	1.613 (1.087–2.394)	0.018	0.039	0.017

Abbreviations: ORs, odds ratios; CIs, confidence intervals. Model 1: adjusted for age, sex, body mass index, and dietary calcium/phosphate intake ratio. Model 2: adjusted for the variables used in Model 1 plus regular exercise, current smoking status, social drinking status, monthly household income, and employed status. Model 3: adjusted for the variables used in Model 2 plus serum vitamin D level.

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
