# Peer review of "Positive Association of Serum Alkaline Phosphatase Level with Severe Knee Osteoarthritis: A Nationwide Population-Based Study"

_diagnostics, 2020, doi:10.3390/diagnostics10121016_

Round 1

Reviewer 1 Report

The present study investigates to what extent an association between elevated serum ALP and severe Knei's arthritis exists. The work could show that a correlation exists. The results are well presented and there are no major points of criticism. The following minor points should be mentioned. Table 2 shows the level of ALP in relation to the extent of OA. However, it would also be interesting to see how the radiological findings are divided into the individual ALP groups (see Table 1). It would also be interesting to see how BMI etc. are distributed in relation to the severity of OA, i.e. a corresponding table for Table 1, but in relation to OA would be helpful.
As far as the discussion is concerned, the question remains open for me as to what the benefit of knowing this correlation is, since I do not assume that high ALP causes osteoarthritis. This reverse conclusion would be wrong for me.

Author Response

<<Reviewer 1>>

The present study investigates to what extent an association between elevated serum ALP and severe Knei's arthritis exists. The work could show that a correlation exists. The results are well presented and there are no major points of criticism. The following minor points should be mentioned.

#1. Table 2 shows the level of ALP in relation to the extent of OA. However, it would also be interesting to see how the radiological findings are divided into the individual ALP groups (see Table 1).

Response: Thank you for your valuable suggestion. We have added the proportion of participants classified by the severity of osteoarthritis according to the tertile of log-transformed ALP level in Table 1 and result section in the revised manuscript as below:

  1. Results

3.1. Clinical characteristics of the study population

The proportion of male sex, social drinker, moderate OA, and severe OA increased with increasing log-transformed serum ALP tertiles.

Table 1. Clinical characteristics of the study population

2008–2011 KNHANES

log-transformed serum ALP level

T1 (1.74–2.32)

T2 (2.33–2.43)

T3 (2.44–3.01)

Total

p

Severity of OA, % (SE)

0.002

Probable OA

46.4 (1.9)

47.8 (2.0)

39.4 (2.1)

44.5 (1.2)

Mild OA

26.9 (1.8)

23.8 (1.7)

25.5 (1.7)

25.5 (1.1)

Moderate OA

19.2 (1.3)

20.8 (1.4)

23.0 (1.6)

21.0 (0.9)

Severe OA

7.5 (1.0)

7.6 (0.9)

12.0 (1.3)

9.1 (0.7)

#2. It would also be interesting to see how BMI etc. are distributed in relation to the severity of OA, i.e. a corresponding table for Table 1, but in relation to OA would be helpful.

Response: We appreciate to your suggestion. The proportion of obese participants and the mean value of BMI increased as the severity of osteoarthritis worsened. We have added Figure S1 and Figure S2 (I have attached Figure S1 and Figure S2 in the word file). We also revised result section of the manuscript as follows:

  1. Results

3.1. Clinical characteristics of the study population

Furthermore, as the severity of osteoarthritis worsened, percentage of obese participants and the mean value of BMI increased (Figure S1 and Figure S2).

#3. As far as the discussion is concerned, the question remains open for me as to what the benefit of knowing this correlation is, since I do not assume that high ALP causes osteoarthritis. This reverse conclusion would be wrong for me.

Response: Thank you for your insightful comments. We also agree that high serum ALP level doesn’t seem to be the cause of osteoarthritis. We assume elevated serum ALP level is an epiphenomenon of inflammation. We have inserted the sentence in discussion section as following.

  1. Discussion

In addition, increased ALP expression can be a cellular reaction to inflammatory stimuli through purinergic signaling which involves chronic inflammation. ALP plays a role in catalyzing pro-inflammatory adenosine triphosphate to anti-inflammatory adenosine. In this context, it is more likely that serum ALP is an epiphenomenon of inflammation rather than the cause of osteoarthritis [26].

In addition, we deleted the following sentences.

This was a cross-sectional study; thus, the causality between serum ALP levels and osteoarthritis could not be determined. The results may indicate a switched causation and a bilateral relationship between serum ALP level and osteoarthritis. Therefore, additional prospective trials with long-term follow-up are also required to establish a causal relationship between ALP level and osteoarthritis progression.

Reviewer 2 Report

Overall this is a well written paper which needs some minor corrections. The authors did a really good job describing their results, but as the readers prefer short papers I would recommend removing some segments of the text. Also the description of results needs a bit of clarification, and some limitations should be mentioned.

Specific comments :

Lines 72-75 – please consider shortening this section, as the diagram following it is self-explanatory

Line 92-93 – please provide an explanation why such stratification of values was chosen

Line 96 - 106 – please clarify if these were AP standing radiographs, or perhaps conventional Xrays taken while patients were in supine position; (extremely important for qualifying patients to various surgical procedures) this could potentially affect inclusion of cases into KL group 0/1, perhaps 2; also consider shortening this segment – the Kellgren – Lawrence scale is well known to readers interested in this topic so a literature reference should be sufficient

117 – inclusion of people who smoked 100+ ciggaretes in their life as current smokers is controversial; (ex. Some patients might have smoked as teenagers but grew-up and quit some decades ago; the effect in these individuals would be negligible); please support it with literature and include this part in the discussion

Line 157 – the sentence regarding the effect of income on ALP – I am not sure what do you mean ; please clarify so that it corresponds with data from the table ; also please include the information that the income was evaluated in materials

Line 164 – the mean BMI …. Did not show statistical significance – I think what you meant is there were no statistically significant DIFFERENCES – please clarify the sentence

Fig.2 – please consider increasing the font size of the text – especially the data near whiskers – the numbers are really tiny ; same applies to Fig. 3

Lines 247-253 – the statements on the relationship between knee pain/circadian disruption/ALP are speculative and unrelated to the study. Although what you wrote is possible, one could expect elevated levels of ALP in all kinds of patients with chronic pain (ex. Low back pain). Please remove or at least tone down these statements.

Limitations – there are some limitations of the study not mentioned in the paper

  1. Did you examine / ask the patients for AO in other joints – hip/shoulder etc. ? Presence of OA in other joints could affect ALP levels
  2. Did you verify patients for degenerative disc disease typically associated with arthritis in intervertebral joints – soma papers suggest that this is the most common location of AO – this could also influence ALP levels
  3. The KL scale … well frankly a lot of orthopaedic surgeons consider it rubbish (and prefer the Ahlback scale), while many authors admit that it is not very precise in grading OA; please discuss a bit on the limitations of radiographic scoring systems, and if you included Xrays taken in supine position – this is also a limitation. From my personal experience there is a high percentage of patients who could be classified as KL 1-2 based in supine Xrays, while standing Xrays, or even better – knee Xrays in Rosenberg view indicate severe OA in the medial compartment with bone on bone contact; a lot of these patients qualify for a UNI knee replacement – this is also confirmed by multiple papers. All in all – if you rely on supine X-rays you will underdiagnose knee OA.
  4. What about pat – fem joint ? Did you include lateral view ? Again – there is a large population of patients with patellar OA who typically are fairly unsypmtomatic – please include this in the discussion
  5. Last – but not least – smoking – please discuss the part regarding 100+ cigarettes – esp. patients who quit a long time ago

Author Response

<<Reviewer 2>>

Overall this is a well written paper which needs some minor corrections. The authors did a really good job describing their results, but as the readers prefer short papers I would recommend removing some segments of the text. Also the description of results needs a bit of clarification, and some limitations should be mentioned.

Response: We thank you for your careful review.

Specific comments:

#1. Lines 72-75 – please consider shortening this section, as the diagram following it is self-explanatory

Response: We have shortened the section from 86 words to 71 words in accordance with Reviewer 2’s comment as follows.

  1. Materials and Methods

2.1. Study population

we applied the following exclusion criteria: 1) no abnormal findings on knee radiographs (n = 2,792), 2) missing data on serum ALP level (n = 403),3) positive for hepatitis B viral surface antigen (n = 149), 4) heavy alcohol consumer (n = 211), 5) patients with rheumatoid arthritis (n = 147), 6) patients with osteoporosis (n = 456), or 7) presence of osteoporosis on dual energy X-ray absorptiometry (n = 541).

#2. Line 92-93 – please provide an explanation why such stratification of values was chosen

Response: Since a total of 3060 people were analyzed, we divided participants into 3 groups according to the tertiles of log-transformed serum ALP level to allocate around 1000 people per group.

#3. Line 96 - 106 – please clarify if these were AP standing radiographs, or perhaps conventional Xrays taken while patients were in supine position; (extremely important for qualifying patients to various surgical procedures) this could potentially affect inclusion of cases into KL group 0/1, perhaps 2; also consider shortening this segment – the Kellgren – Lawrence scale is well known to readers interested in this topic so a literature reference should be sufficient

Response: We totally agree with Reviewer 2’s valuable comment. In the KNHANES, bilateral weight-bearing anteroposterior and lateral X-ray of both knees were obtained. We have revised the Materials and Methods section as follows:

  1. Materials and Methods

2.3. Assessment of knee osteoarthritis

The participants underwent bilateral standing anteroposterior and lateral radiography of both knees using a SD3000 Synchro Stand (Accele Ray Shinyoung Co., Seoul, Korea).

#4. 117 – inclusion of people who smoked 100+ ciggaretes in their life as current smokers is controversial; (ex. Some patients might have smoked as teenagers but grew-up and quit some decades ago; the effect in these individuals would be negligible); please support it with literature and include this part in the discussion

Response: We admit the definition of a current smoker in this study was somewhat confusing. We have changed the Materials and Methods section as below:

  1. Materials and Methods

2.4. Covariate

Among participants who responded that they smoke currently, we defined those who had smoked at least 100 cigarettes during their lifetime as current smokers [17].

#5. Line 157 – the sentence regarding the effect of income on ALP – I am not sure what do you mean ; please clarify so that it corresponds with data from the table ; also please include the information that the income was evaluated in materials

Response: In accordance with Reviewer 2’s comment, we have also added the information how monthly household income was calculated in the Materials and Methods section of the revised manuscript.

  1. Results

3.1. Clinical characteristics of the study population

The proportion of the lowest quartile of monthly household income gradually increased as log-transformed serum ALP tertiles increased. On the other hand, the proportion of the highest quartile of monthly household income gradually decreased as log-transformed serum ALP tertiles increased.

  1. Materials and Methods

2.4. Covariate

Equivalized income was used to represent income considering household size. Monthly household income was calculated by dividing the self-reported monthly household income by the square root of the number of household members. Then, monthly household income was classified into quartiles (lowest, mid-lower, mid-higher, and highest) [20].

#6. Line 164 – the mean BMI …. Did not show statistical significance – I think what you meant is there were no statistically significant DIFFERENCES – please clarify the sentence

Response: I have changed the sentence in accordance with Reviewer 2’s suggestion.

  1. Results

3.1. Clinical characteristics of the study population

Similar trends but with no statistically significant differences were observed in the proportion of current smokers, regular exercise, employed status, and knee joint pain among the groups.

#7. Fig.2 – please consider increasing the font size of the text – especially the data near whiskers – the numbers are really tiny ; same applies to Fig. 3

Response: I have increased the font size of Figure 2. and Figure 3 from 9 to 14.

Figure 2. Comparison of the mean value of the log-transformed serum ALP levels according to (a) osteoarthritis (OA) severity or (b) between severe and non-severe OA.

Figure 3. Percentage of knee pain scale scores according to the tertiles of log-transformed ALP levels.

(I have attached revised Figure 2 and Figure 3 in the word file.)

#8. Lines 247-253 – the statements on the relationship between knee pain/circadian disruption/ALP are speculative and unrelated to the study. Although what you wrote is possible, one could expect elevated levels of ALP in all kinds of patients with chronic pain (ex. Low back pain). Please remove or at least tone down these statements.

Response: Thank you for your thoughtful comments. We have changed the sentences in discussion section as following:

  1. Discussion

Second, elevated serum ALP could be associated with chronic pain. Kamimura el al. found that serum bone specific ALP were remarkably increased in elderly women with lower back pain [31]. This study showed that the participants with severe knee osteoarthritis were more likely to experience pain. Pain induced chronodisruption could be at least partly associated with severity of osteoarthritis. It is well established that persistent pain can provoke sleep disturbance, which is directly connected to alteration of the circadian rhythm. Inversely, sleep disruption can also alter pain processing and lead to a higher prevalence of pain through sensitization and habituation [32]. Chronodisruption could be linked to a substantial increase in inflammatory mediators such as CRP and tumor necrosis factor-α levels, which were increased in osteoarthritic tissue [33-35]. As mentioned earlier, elevated serum ALP levels were definitely associated with increased inflammatory mediators such as CRP level and leukocyte counts [9,25]. In addition, Berenbaum et al. reported that metabolic dysregulation, inflammatory pathways, and the daily rhythmic environment have all been connected to osteoarthritis susceptibility through neuroendocrine systems such as cortisol and parathyroid hormone [36].

#9. Limitations – there are some limitations of the study not mentioned in the paper

#9-1) Did you examine / ask the patients for AO in other joints – hip/shoulder etc. ? Presence of OA in other joints could affect ALP levels

Response: We could not assess osteoarthritis in other joints except knee and hip, because we used secondary data from the KNHANES. The prevalence of hip OA was significantly low compared to knee OA. Moreover, there were no statistically significant differences in the serum ALP level according to the presence or severity of hip OA. We have revised the limitation section.

Table 1 for Reviewers. Prevalence of hip OA according to the severity of knee OA.

Probable knee OA

Mild

knee OA

Moderate knee OA

Severe knee OA

Total

p

0.387

Normal

68.1 (1.7)

64.3 (2.,2)

66.6 (2.4)

69.9 (2.8)

67.0 (1.3)

Probable hip OA

31.2 (1.7)

34.3 (2.2)

31.9 (2.4)

29.9 (2.8)

32.0 (1.3)

Mild hip OA

0.4 (0.2)

1.2 (0.5)

1.3 (0.5)

0.2 (0.2)

0.8 (0.2)

Moderate hip OA

0.3 (0.2)

0.2 (0.2)

0.1 (0.1)

N/A

0.2 (0.1)

Figure 1 for Reviewers. mean value of serum ALP level according to the presence of hip osteoarthritis

Figure 2 for Reviewers. mean value of serum ALP level according to the severity of hip osteoarthritis. 

(I have attached Figure 1 for Reviewers and Figure 2 for Reviewers in the word file.)

  1. Discussion

This study has some limitations that require explanation. We could not assess degenerative disc disease, fat mass, skeletal muscle mass, synovial bone-specific ALP level, and other inflammatory biomarkers, including adipokines and chemokines due to lack of data. Also, there was no information about patellar OA in the KNHANES. In addition, because participants only checked X-ray on knees and hips in the KNHANES, presence of OA in other joints could affect ALP levels although there were no significant differences in serum ALP level according to the presence or severity of hip OA (data not shown). Finally, we could not evaluate knee OA using the Ahlbäck classification. Despite these limitations, our findings, obtained using a complex sampling analysis, can be representative of the actual condition of older Korean adults with osteoarthritis.

#9-2) Did you verify patients for degenerative disc disease typically associated with arthritis in intervertebral joints – soma papers suggest that this is the most common location of AO – this could also influence ALP levels

Response: Unfortunately, we could not verify degenerative disc disease, because we used secondary data from the KNHANES. We have added to the following limitations of the Discussion section.

  1. Discussion

This study has some limitations that require explanation. We could not assess degenerative disc disease, fat mass, skeletal muscle mass, synovial bone-specific ALP level, and other inflammatory biomarkers, including adipokines and chemokines due to lack of data. Also, there was no information about patellar OA in the KNHANES. In addition, because participants only checked X-ray on knees and hips in the KNHANES, presence of OA in other joints could affect ALP levels although there were no significant differences in serum ALP level according to the presence or severity of hip OA (data not shown). Finally, we could not evaluate knee OA using the Ahlbäck classification. Despite these potential limitations, our findings, using a complex sampling analysis, could be representative of the actual condition of older Korean adults with osteoarthritis.

#9-3) The KL scale … well frankly a lot of orthopaedic surgeons consider it rubbish (and prefer the Ahlback scale), while many authors admit that it is not very precise in grading OA; please discuss a bit on the limitations of radiographic scoring systems, and if you included Xrays taken in supine position – this is also a limitation. From my personal experience there is a high percentage of patients who could be classified as KL 1-2 based in supine Xrays, while standing Xrays, or even better – knee Xrays in Rosenberg view indicate severe OA in the medial compartment with bone on bone contact; a lot of these patients qualify for a UNI knee replacement – this is also confirmed by multiple papers. All in all – if you rely on supine X-rays you will underdiagnose knee OA.

Response: We could not get the information about Ahlbäck scale from KNHANES. All radiographic images were reviewed by two radiologists using the Kellgren–Lawrence grading system. Weight-bearing anteroposterior, bilateral standing anteroposterior, and lateral plain radiographs were measured to evaluate the knees. We did not use supine X-ray images. We have inserted following sentences in Method section and limitation section of the revised manuscript.

  1. Materials and Methods

2.3. Assessment of knee osteoarthritis

The participants underwent bilateral standing anteroposterior and lateral radiography of both knees using a SD3000 Synchro Stand (Accele Ray Shinyoung Co., Seoul, Korea).

  1. Discussion

This study has some limitations that require explanation. We could not assess degenerative disc disease, fat mass, skeletal muscle mass, synovial bone-specific ALP level, and other inflammatory biomarkers, including adipokines and chemokines due to lack of data. Also, there was no information about patellar OA in the KNHANES. In addition, because participants only checked X-ray on knees and hips in the KNHANES, presence of OA in other joints could affect ALP levels although there were no significant differences in serum ALP level according to the presence or severity of hip OA (data not shown). Finally, we could not evaluate knee OA using the Ahlbäck classification. Despite these potential limitations, our findings, using a complex sampling analysis, could be representative of the actual condition of older Korean adults with osteoarthritis.

#9-4) What about pat – fem joint ? Did you include lateral view ? Again – there is a large population of patients with patellar OA who typically are fairly unsypmtomatic – please include this in the discussion

Response: Although the participants were taken lateral view X-ray, we could not get information about patellar OA. We could not obtain each official x-ray reading but the presence or absence of Knee OA from the KNHANES data. We have been added to the limitation section of the revised manuscript.

  1. Discussion

This study has some limitations that require explanation. We could not assess degenerative disc disease, fat mass, skeletal muscle mass, synovial bone-specific ALP level, and other inflammatory biomarkers, including adipokines and chemokines due to lack of data. Also, there was no information about patellar OA in the KNHANES. In addition, because participants only checked X-ray on knees and hips in the KNHANES, presence of OA in other joints could affect ALP levels although there were no significant differences in serum ALP level according to the presence or severity of hip OA (data not shown). Finally, we could not evaluate knee OA using the Ahlbäck classification. Despite these potential limitations, our findings, using a complex sampling analysis, could be representative of the actual condition of older Korean adults with osteoarthritis.

#9-5) Last – but not least – smoking – please discuss the part regarding 100+ cigarettes – esp. patients who quit a long time ago

Response: We have inserted the following sentence in Materials and Methods section of the revised manuscript.

  1. Materials and Methods

2.4. Covariate

Among participants who responded that they smoke currently, we defined those who had smoked at least 100 cigarettes during their lifetime as current smokers [17].
